# Review of Template-Based Neuroimaging Tools in Neuro-Oncology: Novel Insights

Jürgen Germann [1], Andrew Yang [1], Clement T. Chow [1], Brendan Santyr [1], Nardin Samuel [1], Artur Vetkas [1], Can Sarica [1], Gavin J. B. Elias [1], Mathew R. Voisin [1], Walter Kucharczyk [2], Gelareh Zadeh [3,4,5,6], Andres M. Lozano [1,6,7] and Alexandre Boutet [2,*]

1. Division of Neurosurgery, Toronto Western Hospital, University of Toronto, Toronto, ON M5T 2S8, Canada
2. Joint Department of Medical Imaging, University of Toronto, Toronto, ON M5T 1W7, Canada
3. Department of Surgery, Division of Neurosurgery, University of Toronto, Toronto, ON M5T2S8, Canada
4. MacFeeters Hamilton Neuro-Oncology Program, Princess Margaret Cancer Centre, Toronto, ON M5G IL7, Canada
5. Princess Margaret Cancer Centre, University Health Network, Toronto, ON M5G 2C1, Canada
6. Krembil Brain Institute, Toronto, ON M5T 2S8, Canada
7. Center for Advancing Neurotechnological Innovation to Application (CRANIA), Toronto, ON M5T 1M8, Canada
* Correspondence: alexandre.boutet@uhn.ca; Tel.: +(416)-603-6200

**Highlights:**

1. Brain MRIs of neuro-oncology patients can be accurately transformed to MNI space.
2. MNI-based studies have provided unique insights to neuro-oncology.
3. While it has proven useful in other fields, it is under-utilized in neuro-oncology.

**Importance of the study**: This is the first review of reference space-based neuro-oncology imaging studies, highlighting their unique scientific contributions from tumor subtyping to symptom mapping and connectomics. However, compared to other fields, reference space-based studies remain few and far between. This neuro-imaging technique has led to high-impact discoveries, for example, in neurodegenerative diseases, suggesting that important research avenues remain untapped, thereby hindering our knowledge of neuro-oncology. Several of these advanced tools and techniques are highlighted in this review with the hope that this will facilitate their implementation in neuro-oncology.

**Abstract**: *Background*: A common MRI reference space allows for easy communication of findings, and has led to high-impact discoveries in neuroscience. Brain MRI of neuro-oncology patients with mass lesions or surgical cavities can now be accurately transformed into reference space, allowing for a reliable comparison across patients. Despite this, it is currently seldom used in neuro-oncology, leaving analytic tools untapped. The aim of this study was to systematically review the neuro-oncology literature utilizing reference space. *Methods*: A systematic review of the neuro-oncology publications was conducted according to PRISMA statement guidelines. Studies specially reporting the use of the Montreal Neurological Institute (MNI) reference space were included. Studies were categorized according to their type of input data and their contributions to the field. A sub-analysis focusing on connectomics and transcriptomics was also included. *Results*: We identified only 101 articles that utilized the MNI brain in neuro-oncology research. Tumor locations ($n = 77$) and direct electrocortical stimulation ($n = 19$) were the most common source of data. A majority of studies ($n = 51$) provided insights on clinical factors such as tumor subtype, growth progression, and prognosis. A small group of studies ($n = 21$) have used the novel connectomic and transcriptomic tools. *Conclusions*: Brain MRI of neuro-oncology patients can be accurately transformed to MNI space. This has contributed to enhance our understanding of a wide variety of clinical questions ranging from tumor subtyping to symptom mapping. Many advanced tools such as connectomics and transcriptomics remain relatively untapped, thereby hindering our knowledge of neuro-oncology.

**Keywords:** magnetic resonance imaging; neuro-oncology; Montreal Neurological Institute brain; MNI152; normalization

## 1. Introduction

Neuroimaging—particularly imaging modalities such as magnetic resonance imaging (MRI)—has long played an indispensable role in neuro-oncology, being crucial for the identification and characterization of tumors and for monitoring treatment response and disease progression [1,2]. Despite this, research in the field of neuro-oncology has yet to fully tap into the potential of template-based neuroimaging methods, as outlined previously [3]. These methods allow detailed spatial mapping at the level of voxels—the small three-dimensional elementary unit constituting an image, equivalent to a pixel in the two-dimensional space of, for instance, a photograph—to characterize, for example, patterns of tumors [3]. Voxel-wise comparisons of MRI findings across multiple patients, possibly with mass lesions or surgical cavities, has historically proven difficult, due to the substantial inter-individual differences in neuroanatomy engendered by their disease process. However, thanks to technological advances, MRI and other neuroimaging modalities of an individual patient's brain can be transformed into a common reference space, which is termed "normalization" allowing for a reliable comparison on a group level. This paradigm shift in neuroimaging research enables more precise quantitative analysis between patients' brains, down to the level of single voxels [3,4].

The most utilized common reference brain space is the Montreal Neurological Institute (MNI) standardized reference space, which was developed in the 1990s [5–8]. The latest version is the MNI152 brain, which was constructed in 2001 from the MRI scans of 152 healthy subjects, and updated in 2009 [3,9]. To effectively use the MNI brain, transforming an individual's brain imaging into the MNI brain space (i.e., normalization) must be as accurate as possible. (Figure S1) Normalization can be performed with clinically acquired high spatial resolution structural MRI. Normalization, the process of transforming a brain to match the template brain, typically involves rotation, translation, scaling, and shearing as a first step. These operations are applied to all voxels in the target image, resulting in 'linear registration'/transformation into MNI space [10]. In the next step, voxels are non-uniformly locally warped to maximize anatomical correspondence between the individual target brain and the template brain. When an individual's anatomy is distorted, it poses additional challenges when normalizing, which can be mitigated with various strategies such as tumor masking [11–13]. Normalization of a brain to a common MNI space allows for a wide variety of uses.

One of the most significant uses of the MNI brain involves applying analyses and tools to characterize specific areas of the brain, and many neuroimaging instruments have been developed for this template. Once individual features of interest and study findings are localized in MNI space, additional analyses can be performed (Figure 1; Table 1). We have termed this approach "spatial characterization analysis" (SCA). SCA makes use of the ever-growing number of spatial reference atlases and maps available. These reference brain atlases and maps are provided in MNI space, and enable scientists to interrogate their respective study findings and identify potential correlates such as molecular and histological markers or developmental processes [14–16]. They include detailed anatomical atlases and maps of regions such as the hypothalamus, thalamic sub-structures and detailed high-resolution molecular maps, clinical atlases, such as function maps that match brain location with capacities including language and motor movement, and maps of brain function derived from large numbers of reported fMRI studies [17–20]. They also include connectomics and transcriptomics maps, enabling brain-wide networks and potential oncogenic biological pathways analysis [21–28]. Using these high-resolution molecular, structural, and functional brain atlases and maps allows the investigator to gain further

insight into the potential molecular, anatomical, and clinical characteristics associated with the brain areas and networks identified in their research.

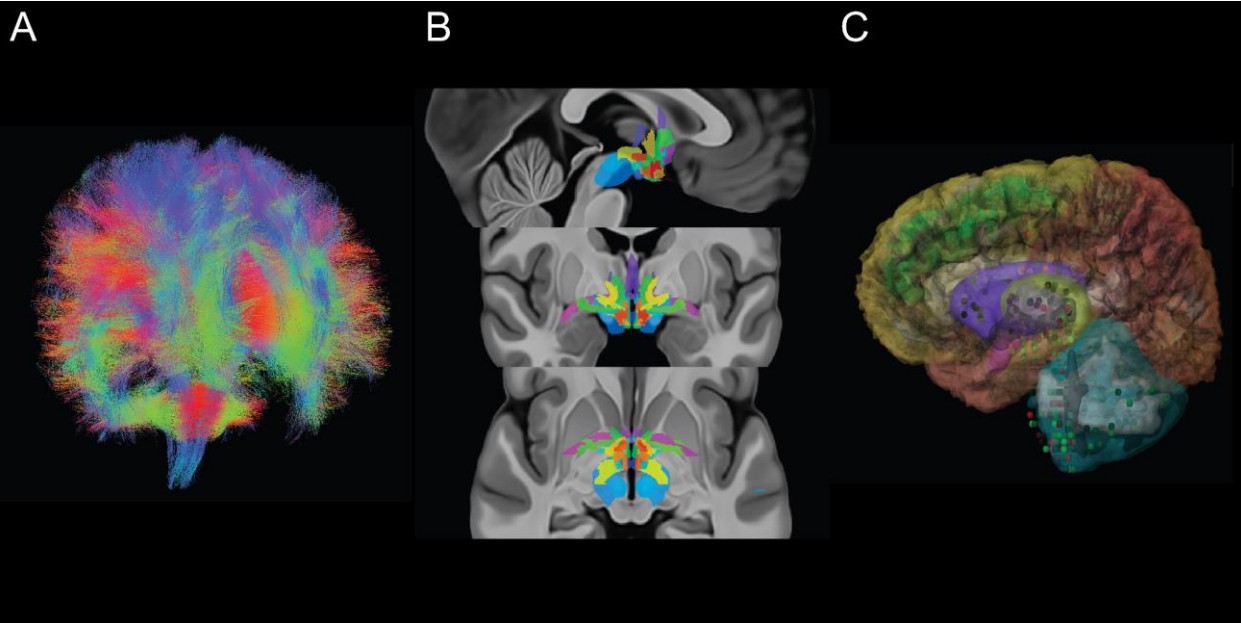

**Figure 1.** Examples of reference brain atlases in MNI space. (**A**) Connectome tractogram comprising ~12 million streamlines derived from normative Human Connectome Project dMRI data [29]. (**B**) High-resolution atlas consisting of manually segmented hypothalamic and extrahypothalamic nuclei [18]. (**C**) Allen Human Brain Atlas, showing gene expression probes for O-6methylguanine-DNA methyltransferse (MGMT) for one of the patients [27].

**Table 1.** Example analysis tools and spatial reference maps available in MNI space.

| Example Analysis Tool | Description | Source |
|---|---|---|
| Functional connectome analysis | Using functional connectivity maps created using fMRI data from a large cohort of healthy individuals (Brain Genomics Superstructure Project dataset; http://neuroinformatics.harvard.edu/gsp; accessed on 1 August 2022) to identify functional brain networks. | http://neuroinformatics.harvard.edu/gsp; accessed on 1 August 2022 https://www.lead-dbs.org/; accessed on 1 August 2022 |
| Structural connectome analysis | Using structural maps constructed using dMRI data from large cohorts of healthy individuals (Human Connectome Project; https://www.humanconnectome.org/; accessed on 1 August 2022) to identify structural brain networks. | https://www.humanconnectome.org/; accessed on 1 August 2022 https://www.lead-dbs.org/; accessed on 1 August 2022 |
| Spatial transcriptome analysis | Using 3D spatial maps of gene expression to identify | https://human.brain-map.org/; accessed on 1 August 2022 https://doi.org/10.7554/eLife.72129; accessed on 1 August 2022 https://abagen.readthedocs.io/en/stable/; accessed on 1 August 2022 |
| High resolution histological atlases | 3D histological brain atlases and architectonic maps (cytoarchitecture and molecular) | https://julich-brain-atlas.de/atlas; accessed on 1 August 2022 https://bigbrainproject.org/; accessed on 1 August 2022 |

**Table 1.** *Cont.*

| Example Analysis Tool | Description | Source |
|---|---|---|
| Hypothalamic atlas | Detailed 3-D atlas of the human hypothalamic regions | https://doi.org/10.5281/zenodo.3724192; accessed on 1 August 2022 https://zenodo.org/record/3724192; accessed on 1 August 2022 |
| Neurosynth maps | Maps of brain areas associated with the function/process of interest, generated using meta analysis of reported fMRI findings | https://neurosynth.org/; accessed on 1 August 2022 |
| Receptor maps | Maps of neurotransmitters generated from large PET imaging datasets | https://doi.org/10.1038/s41593-022-01186-3; accessed on 1 August 2022 https://github.com/netneurolab/hansen_gene-receptor; accessed on 1 August 2022 https://doi.org/10.1073/pnas.1001229107; accessed on 1 August 2022 |
| Cortical expansion during development and evolution | Map of estimated brain expansion during development and evolution | https://doi.org/10.1038/s41592-022-01625-w; accessed on 1 August 2022 https://github.com/netneurolab/neuromaps; accessed on 1 August 2022 |

Abbreviations: MNI: Montreal Neurological Institute; dMRI: diffusion weighted magnetic resonance imaging; fMRI: functional magnetic resonance imaging.

MNI-based neuroimaging research has contributed to unique knowledge in other fields, such as characterizing the addiction network and elucidating neural substrates responsible for clinical benefits in the surgical treatment of obsessive-compulsive disorder [30–32]. As MRI technologies become more refined and readily available in the current era of big data research, it is inevitable that imaging-based neuro-oncology research will rely even more on the MNI brain. Yet, despite the paradigm shift set by the MNI brain, it is currently seldom used in neuro-oncology. This is problematic because an important research tool remains untapped, thereby potentially missing important insights into diagnosis, treatment and prognosis [33]. This review aimed to capture the current state of neuroimaging analysis utilizing this standard reference space. We systematically reviewed neuro-oncology publications that utilize the MNI brain, to highlight various input data sources and findings. Finally, we discuss the untapped research potential within this rapidly growing field and outline various tools and analysis pipelines that are available to scientists who utilize this standardized reference space for their research.

## 2. Methods

This systematic review was conducted according to the PRISMA statement guidelines. MEDLINE was searched from 1995 to 5 July 2022 for studies reporting the use of MNI-brain normalization techniques in neuro-oncology. A combination of Medical Subject Heading (MeSH) terms, synonyms, and keywords for MNI standard space and neuro-oncology was used to maximize sensitivity. Zotero (https://www.zotero.org/, 2022.07.19 Build 6.0.11; accessed on 1 August 2022) was used to collect references, which is a reference manager software [34]. We searched the reference lists of included studies for additional relevant articles. The search strategy is detailed in Table S1. Title and abstract screening, full-text review, and data collection were conducted by two independent reviewers (authors AY and CC), with one other author resolving conflicts (author AB). Covidence (https://www.covidence.org/; accessed on 1 August 2022) was utilized for the systematic review of collected articles [35]. We included original studies reporting the use of the MNI template in neuro-oncology. We excluded studies reporting the use of non-MNI templates such as the SRI24 atlas, non-human subjects, pediatric patients, articles not in English, as well as review articles, commentaries, and editorials.

First, to evaluate the impact, the cumulative publications and number of citations over time were assessed. Second, the input data used to identify areas of the brain for characterization and the contributions to the neuro-oncology field were categorized. This categorization was also conducted specifically for novel connectomic tools. All numerical analyses were conducted using R Studio (https://www.rstudio.com/, 2022.02.03 Build 492) and R (https://www.r-project.org/, accessed on 1 August 2022, version 4.2.1) [36,37].

## 3. Results

We identified 3231 articles through our search of MEDLINE. After 153 duplicates were removed, a total of 3078 articles were identified, which underwent title and abstract screening. A total of 2944 articles were excluded and the full text of 134 articles was reviewed to ensure an optimal selection of studies. After full-text review, 33 additional studies were excluded, mainly due to a lack of use of MNI standard space normalization techniques ($n = 31$). A manual search of reference lists of articles identified no further articles. Overall, 101 studies were included in the final systematic review. Our search is summarized in the Preferred Reporting Items for Systematic Reviews and Meta-Analyses (PRISMA) study flow diagram (Figure 2).

Our search yielded 101 studies with a rapid increase in yearly publications since the first study in 2001 (Figure 3). Median sample size was 100 patients (range: 1–1102) and 50/101 studies had $\geq$100 patients. As of 28 July 2022, the total citations of all studies were 4077 according to CrossRef (https://www.crossref.org/; accessed on 1 August 2022) (Figure S2) [38]. Details for each study are included in Tables S1 and S2.

We first categorized the studies according to the type of input data source, which included tumor location pathology ($n = 77$), direct electrocortical stimulation (DES) mapping ($n = 19$), resection cavity ($n = 3$), infarcts from tumor ($n = 1$), and transcranial magnetic stimulation ($n = 1$) (Figure 4A). Of tumor pathology, the majority were gliomas ($n = 46$), brain metastases ($n = 7$), and meningiomas ($n = 4$). Then, to highlight the contributions of these studies to the field of neuro-oncology, we grouped studies according to their study objectives. Out of the 101 studies, only 4 studies transformed their data in MNI space without further analysis. As this review aimed to highlight various analyses that can be performed after the transformation, we summarized the remainder of the studies in Figure 4B ($n = 97$). Study objectives included correlations with clinical factors ($n = 51$), constructing function maps ($n = 25$), constructing symptom maps ($n = 13$), proposing a new methodology ($n = 14$), and miscellaneous ($n = 10$), such as creating a perfusion map ($n = 3$) and developing a classification system of tumors based on radiomic features ($n = 1$).

As outlined previously, MNI space allows for additional analysis, such as connectomics, transcriptomics and SCA, to be performed. The utilization of high-resolution molecular, structural, and functional brain maps allows researchers to gain further insight into the potential molecular, anatomical, and mechanistic underpinnings associated with the brain areas and networks identified. This is an emerging and powerful approach in the field of neuroimaging; therefore, we focused on these studies and categorized them into the type of input data and their objectives or outputs. Advanced analyses utilized included connectomics ($n = 21$) and transcriptomics ($n = 2$). Connectomics were applied to the studies of tumor lesions ($n = 13$), DES ($n = 7$), and tumor resection cavity ($n = 1$) (Figure 5). The median number of patients for studies was $n = 30$ and the total number of citations was $n = 990$. Connectomic study outputs included correlations with clinical factors ($n = 2$), constructing function maps ($n = 7$), constructing symptom maps ($n = 2$), proposing a new methodology ($n = 6$), and miscellaneous ($n = 4$).

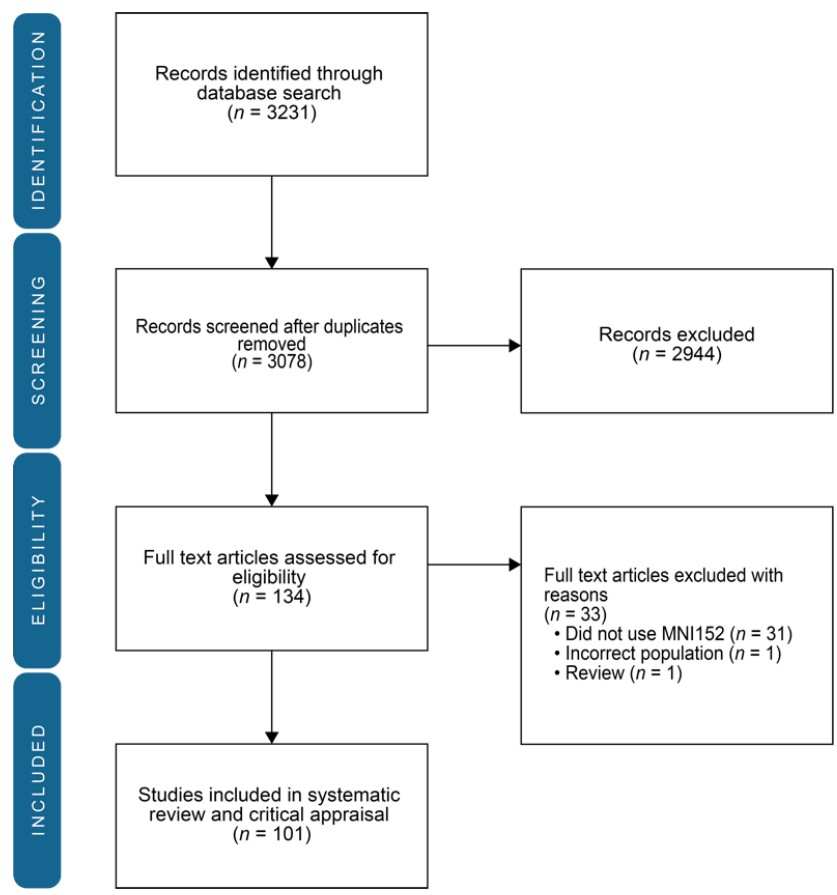

**Figure 2.** PRISMA study flow diagram. Summary of the number of articles screened and included.

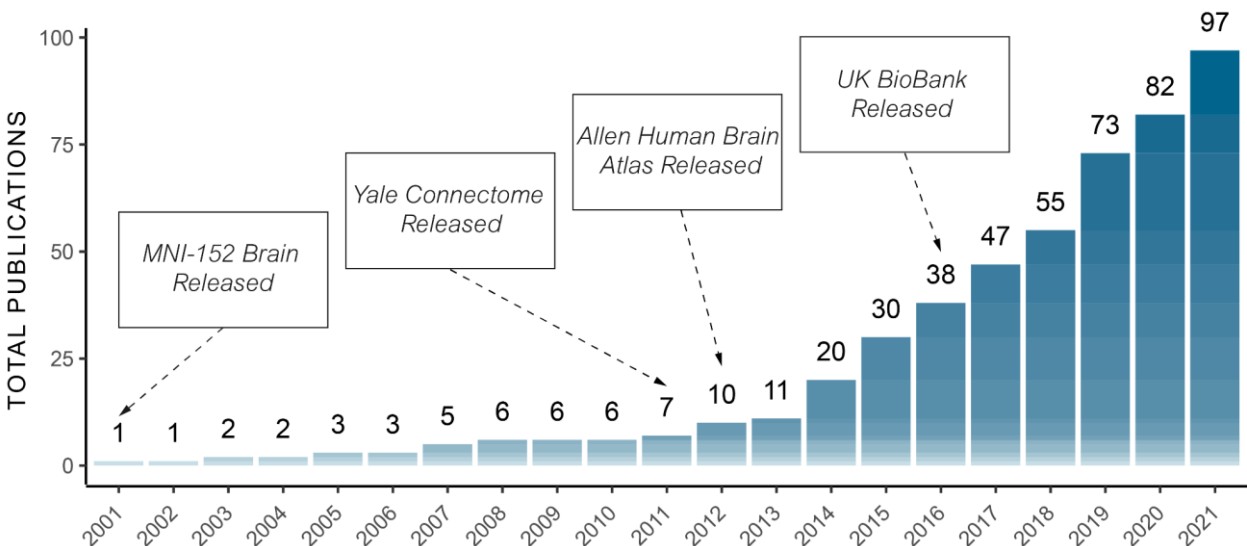

**Figure 3.** Publication trends of the use of MNI space in neuro-oncology research. Each bar represents cumulative publications per year from 2001 to 2021. MNI = Montreal Neurological Institute.

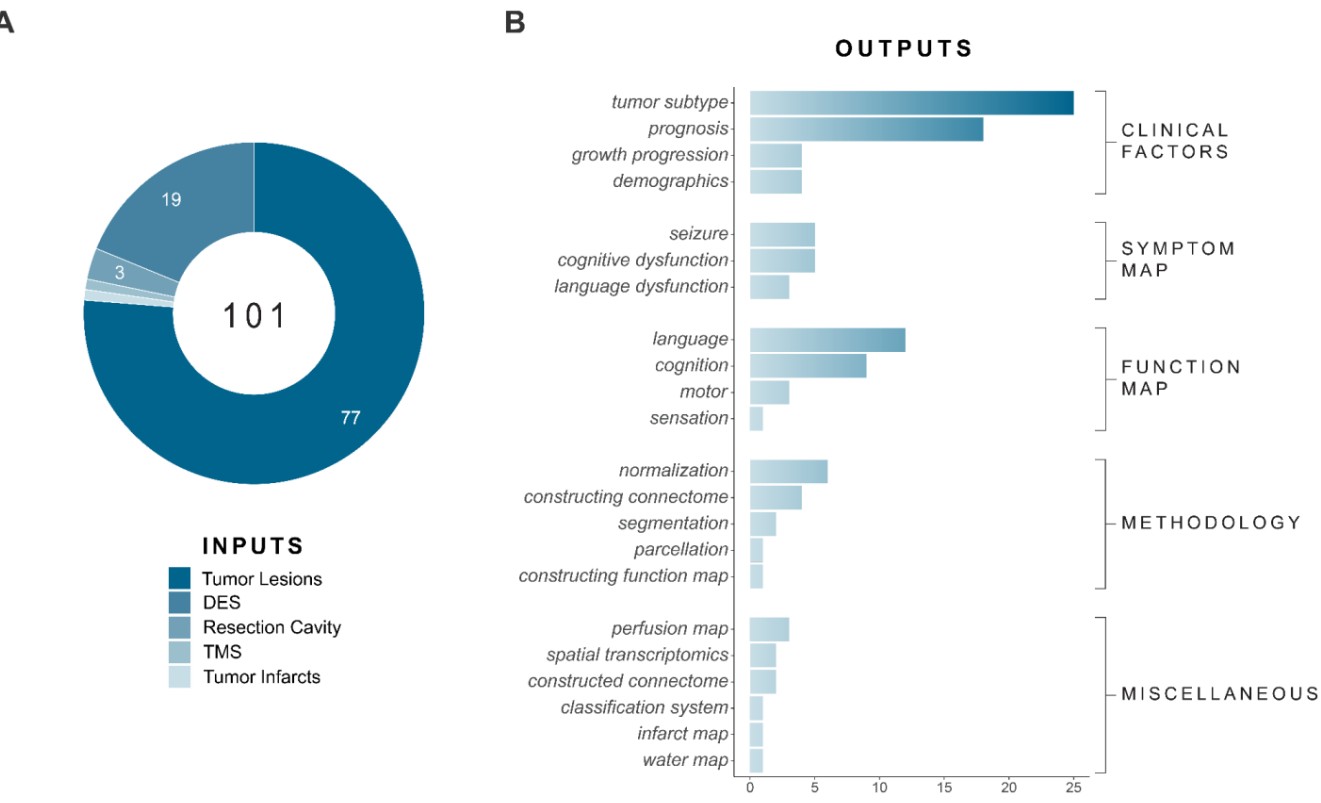

**Figure 4.** Input data and output objectives of all studies. (**A**) The plot indicates the type of input data used to locate an area in the brain for analysis, where tumor lesions were the most common source of data. (**B**) The right plot denotes the study objectives, where the most common was exploring clinical factors. If a study had more than one objective, it was counted more than once. DES = direct electrocortical stimulation; TMS = transcranial magnetic stimulation.

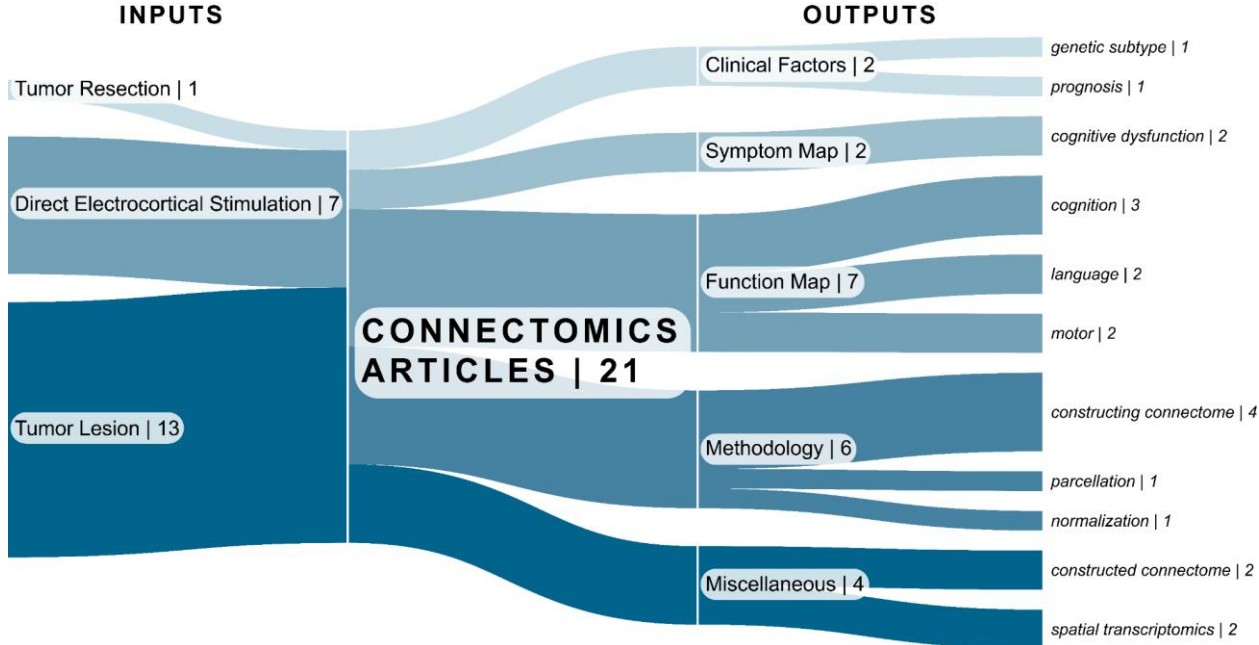

**Figure 5.** Sankey diagram of studies that utilized connectomics, representing the flow of input data sources and output objectives. The left side of the diagram indicates study input data sources and the right side denotes study data output objective.

## 4. Discussion

In this systematic review, we identified 101 articles that utilized the MNI brain in neuro-oncology research. This number is small relative to other fields, such as neurodegenerative disorders, where open access datasets alone have been cited thousands of times [39,40]. Although this is a limited number of articles in the grand scope of neuro-oncology research, we found that an increasing number of neuro-oncology publications are utilizing the MNI brain, which seems to be related to the development of tools, such as the Yale brain connectome (2011), Allen Human Brain Atlas (2012), and the UK Biobank (2016) (Figure 2) [20,24,41]. The novel insights provided by these publications span a wide range, from identifying the spatial distribution of tumor subtypes for enhanced diagnosis to defining white matter networks underlying specific symptoms and functions to creating individualized eloquence maps to guide surgical resection [42,43].

Several untapped research avenues in neuroimaging research in neuro-oncology remain. Only a small number of studies integrated neuroimaging with advanced analysis: 21 publications utilized connectomics, and only two utilized spatial transcriptomics. Connectomic analysis is performed with normative datasets generated using a large dataset acquired with state-of-the-art MRI hardware and software [20,29]. Our review identified a recent study that has used the functional connectomic template as a pre-operative tool to identify regions critical to postoperative cognitive function that should be preserved. Visualizing connections crucial for global brain network function that are at risk from tumor resection can be used to optimize surgical approaches and patient outcomes [44]. Additionally, the structural connectomic template was used to show that white matter properties may explain the preferential location of low-grade gliomas [45]. Both the functional and structural templates can be combined in a multi-modal analysis defining the altered connectomic profiles of glioma patients, which could be used as a future biomarker in this patient population [46]. Spatial transcriptomics is a novel tool that makes use of the Allen Human Brain Atlas. This atlas provides spatially resolved gene expression across 3702 distinct sites in a standardized template space derived from post-mortem tissue samples from six donors [24,25,27]. Spatial transcriptomics could potentially represent a powerful tool to investigate the underlying parenchymal gene expression for various tumor types. Hypothetically, the results could explain why certain brain regions are vulnerable to tumorigenesis and identify new genetic therapeutic targets.

A variety of input data to locate areas of the brain to analyze its characteristics can be used for MNI studies, such as tumor lesions, direct electrocortical stimulation, and resection cavities. For neuro-oncology studies, the most common input data used were tumors. Once the tumors have been transformed into MNI space, a spatial probability map is generated by voxel-wise comparison across patients. Not surprisingly, only four studies in this review reported just the tumor spatial distribution without further analysis. Most studies, once a precise voxel-wise probability map was constructed from input data, performed additional analyses, such as the correlation of spatial location with clinical factors. Clinical factors such as tumor subtyping were most commonly studied. For example, glioblastomas with high VEGF expression are more likely to localize in the left frontal lobe and the right caudate [47]. Similarly, other groups have studied the distribution of different breast metastases subtypes and found that triple-negative metastases occurred more often in the frontal lobe, limbic region, and parietal lobe [48]. These new findings would be amenable to improving diagnosis and predicting tumor type. Moreover, these findings could be used in planning glioma operations and clinical survival predictions [49]. In addition, other types of data, for example, surgical cavities, can be used to explore a drastically different study hypothesis. Tumor resection cavities have been transformed into MNI space to investigate the anatomical culprits for post-operative depression, transient aphasia, as well as behavioral and emotional regulation deficits [43,50,51]. Following further validation, these findings would be useful to guide the extent of resection, prognosticate post-operative outcomes, and more generally understand the relationship between anatomy and function. Interestingly, one group utilized the spatial distribution of infarcts following glioma surgeries,

showing that semantic fluency scores decrease when they occur in the callosal, prefrontal, insulo-opercular, parietal, and temporal deep white matter [52]. Finally, intraoperative DES data have been used for precise mapping of brain function, serving as the basis of functional cortical and subcortical atlases that may be used for pre-surgical planning [53,54]. Various types of input data are amenable to MNI-based analysis, and, depending on the study goals, the most appropriate neuroimaging tool can then be used.

Accuracy of the transformation of individual brains to MNI space is important, and the presence of tumors and/or lesions poses additional challenges. From our review, we identified several methodological studies that attempted to address this concern. Although one study has shown that linear transformation suffices to transform glioma locations in anatomical reference space, non-linear algorithms, which are included in most neuroimaging software, are more accurate [11,55]. Several strategies have been developed to ensure the accuracy of normalization when tumors are present. One approach is to remove the lesion area from the normalization, such that the distortion does not confound the algorithm [53]. Other groups have instead focused on strategies for replacing the tumorous brain with healthy tissue, which can be improved using machine learning [12,13,56]. Alternatively, novel algorithms that identify and exclude pathological regions from being considered during registration can be used to recover tumor brain images to their normal brain appearance counterparts [57,58]. Generally, this is performed with multiple normalization iterations. These techniques have been applied to precisely identify key white matter tracts in tumor patients for pre-surgical planning [59,60]. Moreover, multi-modality registration has been shown to improve the accuracy of the normalization [61].

This study has limitations. First, we only included studies using the MNI brain as a template. Some studies used other templates such as the SRI24 brain [62,63]. However, the MNI brain is by far the most widely used template in neuroimaging due to its quality and, consequently, nearly all maps and tools are built on it. Second, when extracting the results, our study categorization method was subjective. We grouped studies according to their type of input data and study goals to provide a practical and relevant approach for performing MNI-based studies. Third, such a review is subject to publication bias favoring positive studies. Finally, we limited our inclusion criteria to English language studies only, potentially resulting in the omission of certain studies.

Overall, this work aims to advance an emerging field of neuro-oncology research and promote the usage of the MNI reference brain in neuro-oncology neuroimaging research. Having findings and probabilistic spatial maps in standardized space will allow these findings to be more readily compared, help synthesize findings, and advance the field. For example, this was performed with a highly cited meta-analysis tool for fMRI findings in psychology and neurodegenerative disorders [19,39,40]. It will also allow neuro-oncology findings to be compared to findings from other neuroimaging fields, opening avenues for potential discoveries. Furthermore, a great significance of the MNI brain comes from the opportunity to apply additional analyses and tools, as many neuroimaging instruments have been developed for this template. These additional analyses can be performed with clinically acquired high-spatial-resolution structural MRI without the need for additional advanced neuroimaging sequences, and include connectomics analysis, allowing for the identification of brain-wide networks, and spatial transcriptomics to identify biological pathways implicated.

## 5. Conclusions

This review summarizes the neuro-oncology studies using the standard MNI reference brain. This reference space allows neuroimaging findings to be compared between tumor patients and studies, which has enhanced our understanding of a wide variety of clinical questions ranging from tumor subtyping to symptom mapping. However, while the number of these studies has been increasing over time as new tools become available, several potentially high-impact tools, such as connectomics and spatial transcriptomics,

remain unexplored. Unless these powerful tools are further utilized, important research avenues will remain untapped, thereby hindering our knowledge of neuro-oncology.

**Supplementary Materials:** The following supporting information can be downloaded at: https://www.mdpi.com/article/10.3390/onco3010001/s1, Figure S1: Normalization of brain tumor MRI to the Montreal Neurological Institute (MNI) common space template, Figure S2: Cumulative citations over time; Table S1: MEDLINE search terms, Table S2: Articles that utilize MNI in neuro-oncology research.

**Author Contributions:** All the authors helped in the planning, design, and writing of the review as well as critically reviewed the manuscript and met authorship requirements. Conception and design: J.G., A.B., A.Y.; acquisition of data: A.Y., C.T.C.; analysis and interpretation of data: J.G., A.B., A.Y., C.T.C.; drafting the article: J.G., A.B., A.Y., C.T.C., B.S.; critically revising the article: all authors; reviewed submitted version of manuscript: all authors; administrative/technical/material support: J.G., A.B.; study supervision: J.G., A.B. All authors have read and agreed to the published version of the manuscript.

**Funding:** This work was supported by a Canadian Institutes of Health Research Banting fellowship (#471913) to J.G. (Canada).

**Institutional Review Board Statement:** Not applicable.

**Informed Consent Statement:** Not applicable.

**Data Availability Statement:** The data presented in this study are available upon request from the corresponding author.

**Conflicts of Interest:** A.M.L. is Scientific Director for Functional Neuromodulation Ltd. and a consultant to Medtronic, Abbott, Boston Scientific, Insightec, and Focused Ultrasound Foundation. The remaining authors have no conflict of interest to disclose.

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
