# Peer review of "Review of Template-Based Neuroimaging Tools in Neuro-Oncology: Novel Insights"

_onco, doi:10.3390/onco3010001_

Round 1

Reviewer 1 Report (Previous Reviewer 1)

The supplementary table is not supplied with the manuscript. Moreover, the text lacks reference to the found papers directly where the respective research / methods are addressed.

As this is crucial for a reference  that should collect all evidence from the literature another round of review is required to yield a proper review.

Author Response

Alexandre Boutet

Department of Medical Imaging

University of Toronto

263 McCaul St. Fl 4

Toronto, ON M5T 1W7

Canada

Tel: (416) 603-6200

[email protected]

Prof. Dr. Fred Saad

Editor-In-Chief

Onco

November 29, 2022

Dear Dr. Saad,

Many thanks for giving us the opportunity to submit a second revision of our manuscript “Review of template-based neuroimaging tools in neuro-oncology: novel insights.” (onco-2078493). We appreciate the editors’ and reviewers’ careful consideration of our work and thank them for the thoughtful reviews.

Three files are included with these revisions: a reply letter, the main manuscript, and supplementary materials.

Please find a detailed point by point response to each query. We feel that with these changes the paper is clearer and stronger. The changes to the tables and figures are summarized at the end. The reviewers’ comments are in BOLD and our replies are in plain text. Changes made to the manuscript were tracked.

Reviewer 1:

The supplementary table is not supplied with the manuscript. Moreover, the text lacks reference to the found papers directly where the respective research / methods are addressed.

As this is crucial for a reference  that should collect all evidence from the literature another round of review is required to yield a proper review.

  • Thank you for the suggestions. We apologize that the reviewer could not find the Supplementary Table 2 attached in the previous revision file. It is included with the current manuscript (as a separate document named “Supplementary_Material”). For completeness, we added a supplementary table (Supplementary Table 2) detailing each reference included and how they were categorised. This was specified in the Results section. As mentioned previously, the purpose of the review was to summarise the literature using categorization of the publications (Figure 4) and emphasising promising studies that utilized connectomics (Figure 5). Thus, the discussion was based on those categories referencing key papers in the field. Unfortunately, it would not be possible to cite the 101 references in the discussion as it would probably impair its readability. Therefore, we used the results section with figures to summarize and also included a detailed supplementary table with the 101 included studies (See Supplementary Table 2). Our search was systematic following PRISMA Guidelines (see Methods section) and supplementary table 1.

Tables and Figures summary of changes to accommodate the comments:

  • Table 1: No change.
  • Figure 1: No change.
  • Figure 2: No change.
  • Figure 3: No change.
  • Figure 4: No change.
  • Figure 5: No change.
  • Supplementary Table 1: No change.
  • Supplementary Table 2: No change.
  • Supplementary Figure 1: No change.

Once again, we thank the reviewers and we think the manuscript is stronger and clearer and hope you will now find it suitable for publication. We will add our PROSPERO once we receive our number. We very much look forward to hearing from you.

Sincerely,

Alexandre Boutet, MD, Ph.D.

Joint Department of Medical Imaging

University of Toronto

Round 2

Reviewer 1 Report (Previous Reviewer 1)

The authors now have supplied valuable information in the supplementary material. MeSH terms are provided and a listing of the papers retrieved. This latter list is informative yet lacks proper referencing of the cited publications.

Retrieved references that are addressed in the body of the paper are cited completely. All the other works found should be made available to the readership in a suitable way, too.

Unfortunately the current form is deemed inadequate: writing e.g. "Wang et al", at the paper "Exploration of spatial distribution of brain....." in Supp. Table 1. This table should include the complete reference making a review complete and valuable.  With a modern literature management system this is a fairly easy effort...

In the text the referencing of Supp. Figs 1 and 2 is mixed: on p. 2 Supp Fig 1 is referenced, on p. 5 Supp. Fig. 2 but it should be flipped. The re-labeling the figures in the supplement will correct for this.

Author Response

Reviewer 1:

The authors now have supplied valuable information in the supplementary material. MeSH terms are provided and a listing of the papers retrieved. This latter list is informative yet lacks proper referencing of the cited publications.

  • Thank you for the kind comments.

Retrieved references that are addressed in the body of the paper are cited completely. All the other works found should be made available to the readership in a suitable way, too.

Unfortunately the current form is deemed inadequate: writing e.g. "Wang et al", at the paper "Exploration of spatial distribution of brain....." in Supp. Table 1. This table should include the complete reference making a review complete and valuable.  With a modern literature management system this is a fairly easy effort...

  • Thank you for the suggestions. We have included the full references of the 101 articles that we have included in our manuscript in the supplementary materials document. They are identified with superscript in Supplementary Table 2.

In the text the referencing of Supp. Figs 1 and 2 is mixed: on p. 2 Supp Fig 1 is referenced, on p. 5 Supp. Fig. 2 but it should be flipped. The re-labeling the figures in the supplement will correct for this.

  • We have also fixed the order of the supplementary figures in the supplementary materials document to reflect the correct order found in the manuscript. Thank you for bringing this to our attention.

Tables and Figures summary of changes to accommodate the comments:

  • Table 1: No change.
  • Figure 1: No change.
  • Figure 2: No change.
  • Figure 3: No change.
  • Figure 4: No change.
  • Figure 5: No change.
  • Supplementary Table 1: No change.
  • Supplementary Table 2: Superscripts added to identify the references.
  • Supplementary Figure 1: Changed order to reflect correct order found in manuscript.
  • Supplementary Figure 2: Changed order to reflect correct order found in manuscript.

Once again, we thank the reviewers, and we think the manuscript is stronger and clearer and hope you will now find it suitable for publication. We very much look forward to hearing from you.

Sincerely,

Alexandre Boutet, MD, Ph.D.

Joint Department of Medical Imaging

University of Toronto

This manuscript is a resubmission of an earlier submission. The following is a list of the peer review reports and author responses from that submission.

Round 1

Reviewer 1 Report

This is a nicely written review of the use of the MNI data set in the current publications of the field.

As there is nothing to add methodologically, the direct referencing of the found papers seems to be absent in the manuscript. On certain occasions some papers are referenced and / or discussed. The majority of the papers is not presented.

For a review it would indicated to assign the found papers, i.e. the content of a review, to the contents presented. An ideal location would be to add the papers in Fig. 5 or whenever they are mentioned in the text. In its current form the reader is incompleterly informed by stating that 101 papers were identified but not given. For a review there is naturally ample space for cited literature.

Please consier accordingly.

Reviewer 2 Report

The manuscript, altough it is not of high scientific value and quality, could be considered for publication. First, the Discussion should be shortened and synthetised.

Reviewer 3 Report

Well presented review paper with a lot of information. I don't have many corrections and comments other than these few notes in the methods section.

Methods:

Line: 131 The Zotero website and its contents could possibly be explained more.

Please define these  acronyms.
Line 135:  (AY and CC), 
Line 136: (AB) 
